# Impact of the COVID-19 pandemic on early career dementia researchers: A global online survey

Sara Laureen Bartels[1]*, C. Elizabeth Shaaban[2], Wagner S. Brum[3,4], Lindsay A. Welikovitch[5], Royhaan Folarin[6], Adam Smith[7], on behalf of the ISTAART PIA to Elevate Early Career Researchers¶

1 Department of Psychiatry and Neuropsychology and Alzheimer Centre Limburg, School for Mental Health and Neurosciences, Maastricht University, Maastricht, The Netherlands, 2 Department of Epidemiology, School of Public Health, University of Pittsburgh, Pittsburgh, PA, United States of America, 3 Graduate Program in Biological Sciences: Biochemistry, Universidade Federal do Rio Grande do Sul (UFRGS), Porto Alegre, Brazil, 4 Department of Psychiatry and Neurochemistry, The Sahlgrenska Academy at the University of Gothenburg, Mölndal, Sweden, 5 Department of Neurology, Massachusetts General Hospital and Harvard Medical School, Boston, MA, United States of America, 6 Department of Anatomy, Olabisi Onabanjo University, Sagamu, Nigeria, 7 Institute of Neurology, University College London, London, The United Kingdom

¶ Membership of the Alzheimer's Association International Society to Advance Alzheimer's Research and Treatment (ISTAART) Professional Interest Area (PIA) to Elevate Early Career Researchers (PEERs) is provided in the Acknowledgments.
* sara.bartels@maastrichtuniversity.nl

**Data Availability Statement:** The data that support the findings of this study are openly available in the Open Science Framework: Bartels, S. L. (2022, October 5). "Impact of the COVID-19 Pandemic on

## Abstract

### Introduction

The World Health Organization recognizes dementia as a public health priority and highlights research as an action to respond to the consequences, with early career dementia researchers (ECDRs) representing the key driving force. Due to the COVID-19 pandemic, however, biomedical and psychosocial dementia research was strained worldwide. The aim of this study was to understand the impact of the pandemic on ECDRs.

### Methods

In autumn 2021, the Alzheimer's Association International Society to Advance Alzheimer's Research and Treatment (ISTAART) Professional Interest Area to Elevate Early Career Researchers (PEERs) and University College London conducted an online survey querying ECDRs' experiences during the COVID-19 pandemic. The survey was shared through the ISTAART network, social media, podcasts, and emailing lists. Data were analyzed using descriptive and inferential statistics.

### Results

Survey data from n = 321 ECDRs from 34 countries were analyzed (67.6% women; 78.8% working in academia). Overall, 77.8% of ECDRs surveyed indicated research delays, 53.9% made project adjustments, 37.9% required additional or extended funding, and 41.8%

Early Career Dementia Researchers: A Global
Online Survey". Retrieved from www.osf.io/fr8j2.

**Funding:** No funding was received for this work
specifically. CES was funded by grant
T32AG055381 from the National Institute on Aging
of the United States National Institutes of Health.
AS is funded by the National Institute for Health
Research through the NIHR University College
London Hospitals Biomedical Research Centre.
This manuscript was facilitated by the Alzheimer's
Association International Society to Advance
Alzheimer's Research and Treatment (ISTAART),
through the PEERS PIA. The views and opinions
expressed by authors in this publication represent
those of the authors and do not necessarily reflect
those of the PIA membership, ISTAART, or the
Alzheimer's Association. The funders had no role in
study design, data collection and analysis, decision
to publish, or preparation of the manuscript.

**Competing interests:** The authors have declared
that no competing interests exist.

reported a negative impact on career progression. Moreover, 19.9% felt unsupported by their institutions and employers (33% felt well supported, 42.7% somewhat supported). ECDR's conference attendance remained the same (26.5%) or increased (More: 28.6%; a lot more: 5.6%) since the start of the pandemic. Continental differences were visible, while the impact of the pandemic did not differ greatly based on ECDRs' sociodemographic characteristics.

## Conclusions

The COVID-19 pandemic had a substantial impact on ECDRs worldwide and institutions, employers, and funding bodies are urged to consider the implications and lessons-learned when working with, managing, and promoting ECDRs. Strategies related to the pandemic and general career support to improve ECDRs career progression are discussed, including social media training, digital networking, and benefits of hybrid events. Global resources specific for ECDRs are highlighted.

## Introduction

Researchers across all disciplines are grappling with the consequences of the COVID-19 pandemic, both personally and professionally; however, early career researchers (ECRs) represent a particularly vulnerable demographic in biomedical and psychosocial research. A review of the pertinent literature reveals that ECRs have been disproportionally affected by the COVID-19 pandemic [1]. Compared to researchers at more advanced career stages, ECRs have a less established record of independent funding or publications, deal with a more vulnerable income situation, and are therefore strongly affected by research-related and current circumstantial stressors [2, 3]. There is no universal definition for ECRs due to differences in job titles and career paths across the world. Therefore, the term ECR refers here to researchers at the undergraduate, graduate, postdoctoral, and early career faculty level who identify as being at an early career stage.

Across various research fields, the impact of the pandemic on the academic and personal status of ECRs is being investigated. For instance, a survey of researchers working in Animal Behavior and Welfare, other biological sciences, or social sciences conducted in July 2020 revealed that postgraduate students, research associates, and non-permanent jobholders were more likely to worry about their future compared to permanent jobholders [4]. Amongst a group of Italian neurology trainees, 60% of those surveyed in April 2020 indicated a reduction in research activities, and about 70% feared the negative impact of the pandemic on their training [5]. In the same month, trainees and early career faculty members of the Association for Academic Surgery and the Society of University Surgeons reported a greater negative impact of the pandemic on productivity, and having received less guidance from their institution compared to advanced career faculty counterparts [6]. Furthermore, a survey of the Autism Science Foundation and Autism Speaks presents that 85% of surveyed ECRs experienced reduced productivity, particularly struggling with recruitment of research participants, and dealing with increased needs at home and personal mental health [7].

The COVID-19 pandemic and impact may also have lasting consequences, on ECRs' career progression [8]. A report published in *Nature* presents results from over 7600 multi-disciplinary postdoctoral researchers worldwide, highlighting that 61% of respondents worry about the

pandemic's negative effects on their career prospects [9]. ECRs in cardiovascular research fear reduced opportunities for training, peer learning, networking, conference presentations, and recruitment issues impacting publications of original research, which may affect their career progression [8]. The current increased attention on COVID-19 research projects has changed the landscape of funding bodies, offering novel opportunities for some ECRs, but can also reinforce pre-existing inequities in academia [10].

In dementia research specifically, important progress, for instance in blood-based biomarkers for early detection and prevention of Alzheimer's disease, was achieved despite the pandemic [11]. Research conducted in 2020 and 2021 also investigated the impact of the pandemic on people with dementia and their carers, highlighting severe disruptions of routines that support mental and physical health in these population [12, 13]. Dementia is recognized by the World Health Organization as a public health priority and research is promoted as one of the actions in the Global action plan on the public health response to dementia 2017–2025 [14]. However, investments in the COVID-19 illness are thought to be higher than those dedicated to dementia research and treatment [15], creating not only an issue for people affected by dementia, but also ECRs and their dementia-related work.

For ECRs working in dementia research, a recent call for action urges research and academic institutions, governments, funding agencies, and advanced researchers in the field to invest time and resources in career development [16]. ECRs are the key driving force to advance research into treating and preventing dementia and related conditions, as well as to counteract the impact of the pandemic on older adults living with cognitive impairments due to Alzheimer's disease and other types of dementia. While several of the aforementioned reports provided insight on the effects of the pandemic over ECRs in other research fields, little is known about the impact of the COVID-19 pandemic on early career dementia researchers (ECDRs). The Alzheimer's Association International Society to Advance Alzheimer's Research and Treatment (ISTAART) Professional Interest Area (PIA) to Elevate Early Career Researchers (PEERs), comprising 374 members from 39 countries (status: March 2022) and Continent Lead Executive Committee members from six continents, is in a unique position to engage multi-disciplinary ECDRs worldwide. The aim of the present study was to evaluate the impact of the COVID-19 pandemic on ECDRs, and identify the extent to which diverse work environments supported them during this crisis. Continental trends and differences in ECDR sociodemographic characteristics were also explored. Strategies are presented to stimulate institutions, employers, and individuals to provide ECDRs with better career development opportunities during and beyond the pandemic.

## Materials and methods

### Survey

This study was executed by the ISTAART PIA to Elevate Early Career Researchers (PEERs) and University College London. The survey created in English using www.surveymonkey.com, and covered several topics (e.g., experiences, job and workplace, conference attendance, publishing, moving countries, leaving academia), including the impact of COVID-19. The full survey report (non-peer reviewed) with descriptive information can be accessed elsewhere [17]. Participation was anonymous and individuals could participate in the survey if they self-identified as (pre-tenure) ECRs currently working in any of the multidisciplinary fields in dementia research or if they had left the field within the last two years. The survey could be paused and re-entered via the same web browser at a later time for continuation. No compensation was offered. The survey was distributed via social media, newsletters, podcasts, blogs, e-mails to ISTAART ECDRs, various departments, institutions, networks and charities, and was

available between the 1st of September, 2021, and 31st of October, 2021 (towards the end of the third COVID-19 wave according to the World Health Organization). The study was approved by the University College London Research Ethics Committee REC (Number 21275/001). Via the online platform, participants read the study information and data use regulations, and then confirmed eligibility and provided written informed consent by ticking a box ("I agree"). A name or contact information was not collected. If participants did not confirm eligibility, or did not provide informed consent ("I disagree"), participation ended without collecting any information.

### Statistical analysis

Descriptive statistics are presented as numbers and percentages, and bar graphs were used to visualize results. Inferential statistical tests were carried out with chi-square tests.

## Results

### Sample characteristics

In total, n = 584 ECDRs responded to the survey, and n = 321 ECDRs completed the COVID-19 section (n = 308 currently work in the field of dementia research, n = 13 left the field in the past two years). Socio-demographic (Table 1) and work-related characteristics (Table 2) are presented below.

### Global impact of the COVID-19 pandemic on the research and career of ECDRs

Across all responders, 77.8% reported a delay in their project due to the pandemic, 53.9% had to adjust their research plans, and 37.9% needed to secure a funding extension or additional funding. Furthermore, 41.8% indicated a negative impact of the COVID-19 pandemic on career progression due to a lack of funding or job opportunities. Among respondents, 33% found that their institutions or employers supported them well during the pandemic, while 42.7% indicated that their institution or employers were generally supportive but could have been better, and 19.9% felt unsupported by their institution or employers. Also, 34.2% (28.6% "more", 5.6% "a lot more") of participants reported increased conference attendance, while conference attendance was the same for 26.5% of the surveyed ECDRs since the start of the pandemic. Further details can be found in Fig 1A–1F.

### Continental trends

The data was explored with a focus on continental trends using bar graphs. Inferential statistical analyses were not performed in this section due to small sample sizes in certain regional groups, which should be considered when interpreting the graphs. Delays in research projects were experienced by the majority of ECDRs across all continents. Trends also include, for instance, that 81% of the surveyed ECDRs working in Africa (n = 21) had to make no adjustments to their projects, while 57.1% also did not perceive their institutions as supportive. Conference attendance increased especially in Europe (31.7% "more", 3.3% "a lot more"), Asia (50% "more"), and South/Central America (36.7% "more", 13.3% "a lot more"). Further details can be seen in Fig 2A–2F.

### Gender differences

Gender differences were investigated comparing men and women (not including n = 8 who were genderqueer, nonbinary, or preferred to self-describe due to the small sample size). No

gender differences were present in this sample regarding the perceived impact of the pandemic on any of the categories: delay in research projects, $X^2$ (df = 3, n = 312) = 2.48, p = .48; the need to adjust or rethink research project, $X^2$ (df = 2, n = 313) = 1.26, p = .53; the need to secure an extension or additional funding, $X^2$ (df = 3, n = 311) = 2.65, p = .45; perceived impact on career progression, $X^2$ (df = 3, n = 310) = 2.00, p = .57; perceived support from the

**Table 1. Socio-demographic characteristics of the surveyed ECRs (n = 321).**

| | n | % |
|---|---|---|
| **Age** | | |
| <18–24 | 31 | 9.7 |
| 25–34 | 160 | 49.8 |
| 35–44 | 96 | 29.9 |
| 45–54 | 22 | 6.9 |
| 55+ | 11 | 3.4 |
| Prefer not to answer | 1 | 0.3 |
| **Gender** | | |
| Woman | 217 | 67.6 |
| Man | 96 | 29.9 |
| Genderqueer/ non-binary/ prefer to self-describe | 8 | 2.5 |
| **Nationality** | | |
| American | 58 | 18.1 |
| Argentinian, Belgian, South Korean | 3[1] | 0.9[1] |
| Australian | 8 | 2.5 |
| Brazilian | 25 | 7.8 |
| Canadian, Indian, Irish | 9[1] | 2.8[1] |
| Chinese | 12 | 3.7 |
| Danish, French, Italian, Mexican, Portuguese, Spanish | 4[1] | 1.2[1] |
| Dutch, Nigerian | 18[1] | 5.6[1] |
| English | 49 | 15.3 |
| German | 13 | 4 |
| Ghanaian, Singaporean, Swedish, Swiss | 2[a] | 0.6[a] |
| Scottish | 10 | 3.1 |
| Cameroonian, Chilean, Colombian, Congolese, Costa Rican, Cuban, Cypriot, Czech, Greek, Guyanese, Indonesian, Iranian, Iraqi, Israeli, New Zealander, Norwegian, Peruvian, Polish, Puerto Rican, Rwandan, Salvadorean, Taiwanese, Ugandan, Welsh | 1[a] | 0.3[a] |
| Prefer not to answer | 14 | 4.4 |
| **Treated or perceived as a racial minority or person of color at current residence** | | |
| Yes | 59 | 18.4 |
| No | 255 | 79.4 |
| Prefer not to answer | 7 | 2.2 |
| **Having any dependents under the age of 18** | | |
| Yes | 74 | 23.1 |
| No | 243 | 75.7 |
| Prefer not to answer | 4 | 1.2 |
| **Being a primary caregiver** | | |
| Yes | 69 | 21.5 |
| No | 247 | 76.9 |
| Prefer not to answer | 5 | 1.6 |

[a] These nationalities had the same number of responders and were therefore presented in the same rows.

**Table 2. Work-related characteristics of the surveyed ECRs (n = 321).**

| | n | % |
|---|---|---|
| **Position** | | |
| Undergraduate student | 16 | 5 |
| PhD/ graduate student | 118 | 36.8 |
| Postdoctoral researcher/ research fellow | 100 | 31.2 |
| Assistant professor | 41 | 12.8 |
| Associate/ full professor | 13 | 4.0 |
| Other (e.g., lecturer, instructor, MD psychiatrist, research associate, research scientist, trainee) | 33 | 5 |
| **Research field** (multiple selections possible) | | |
| Arts and dementia | 21 | - |
| Basic sciences and pathogenesis | 95 | |
| Biomarkers | 106 | |
| Clinical | 64 | |
| Communities/ environment | 31 | |
| Data analysis | 82 | |
| Delivery of drug trials | 5 | |
| Dementia care | 74 | |
| Drug discovery/ development | 22 | |
| Neuropsychology | 70 | |
| Patient and public involvement | 35 | |
| Public health | 65 | |
| Social care | 39 | |
| Technology | 29 | |
| Other | 27 | |
| **Working location** | | |
| Africa [a] | 21 | 6.5 |
| Asia [b] | 14 | 4.4 |
| Australia | 11 | 3.4 |
| Central/ South America [c] | 31 | 9.7 |
| Europe [d] | 142 | 44.2 |
| North America [e] | 99 | 30.8 |
| Prefer not to answer | 3 | 0.9 |
| **Funding source of current position** | | |
| Dementia charity | 28 | 8.7 |
| Non-dementia charity | 10 | 3.1 |
| Foundation | 17 | 5.3 |
| Government agency | 104 | 32.4 |
| Not currently working/ studying | 5 | 1.6 |
| Private company/ commercial | 4 | 1.9 |
| Self-funded | 31 | 9.7 |
| University | 92 | 28.7 |
| Other (e.g., European Union, medical council, self-employed) | 28 | 8.7 |
| **Workplace** | | |
| Academia (university or college) | 253 | 78.8 |
| Government | 8 | 2.5 |
| Hospital or clinic | 37 | 11.5 |
| Non-profit organization | 7 | 2.2 |

*(Continued)*

**Table 2.** (Continued)

| | n | % |
|---|---|---|
| Other (e.g., academic hospital, non-university research institute, volunteer, industry, residential care facility) | 11 | 3.4 |
| **Length of contract** | | |
| < 1 year | 26 | 8.1 |
| 1 year | 50 | 15.6 |
| 2 years | 44 | 13.7 |
| 3 years | 66 | 20.6 |
| 4 years | 43 | 13.4 |
| 5 years | 29 | 9.0 |
| Permanent position | 47 | 14.6 |
| Not currently working/ studying | 4 | 1.2 |
| **Time left on contract** | | |
| < 6 months | 52 | 16.2 |
| 6–12 months | 69 | 21.5 |
| 1–3 years | 107 | 33.3 |
| 3–5 years | 20 | 6.2 |
| 5+ years | 5 | 1.6 |
| Permanent position | 46 | 14.3 |

Note regarding "Working location": Respondents came from these countries, specifically:

[a] Cameroon n = 1, Democratic Republic of Congo n = 1, Ghana n = 1, Nigeria n = 16, Rwanda n = 1, Uganda n = 1.

[b] China n = 8, India n = 4, Israel n = 1, Taiwan n = 1.

[c] Argentina n = 3, Brazil n = 24, Chile n = 1, Colombia n = 1, Costa Rica n = 1, El Salvador n = 1

[d] Belgium = 1, Cyprus n = 1, Czech Republic n = 1, Denmark n = 4, France n = 2, Germany n = 11, Ireland n = 4, the Netherlands n = 21, Norway n = 3, Portugal n = 1, Spain n = 2, Sweden n = 10, Switzerland n = 1, United Kingdom n = 80.

[e] United States of America n = 91, Canada n = 6, Mexico n = 2.

institution or employer, $X^2$ (df = 4, n = 313) = 6.33, p = .18; or conference attendance, $X^2$ (df = 4, n = 280) = 1.81, p = .77.

## Racial differences

The frequencies of EDCRs replies who indicated being treated or perceived as a racial minority or person of color (n = 59) were compared to EDCRs who were not treated or perceived as a racial minority or person of color (n = 255) at their current living situation. There were no significant differences between these two groups in the pandemic's impact in any category: research project delays, $X^2$ (df = 3, n = 313) = 1.51, p = .68; changes or adjustments in research projects, $X^2$ (df = 2, n = 314) = 2.51, p = .29; the need to secure for additional funding or an extension, $X^2$ (df = 3, n = 312) = 1.64, p = .65; perceived impact on career progression, $X^2$ (df = 3, n = 311) = 3.0, p = .39; perceived support from institutions or employers, $X^2$ (df = 4, n = 314) = 3.4, p = .50; or conference attendance, $X^2$ (df = 4, n = 281) = 4.8, p = .31.

## Age differences

Age differences were evaluated after combining the seven age groups (Table 1) into four groups (excluding n = 1 who preferred not to answer): Group 1 (n = 31): under 18–24 years, Group 2 (n = 159): 25–34 years., Group 3 (n = 96): 35–44 years, Group 4 (n = 33): 45 years +.

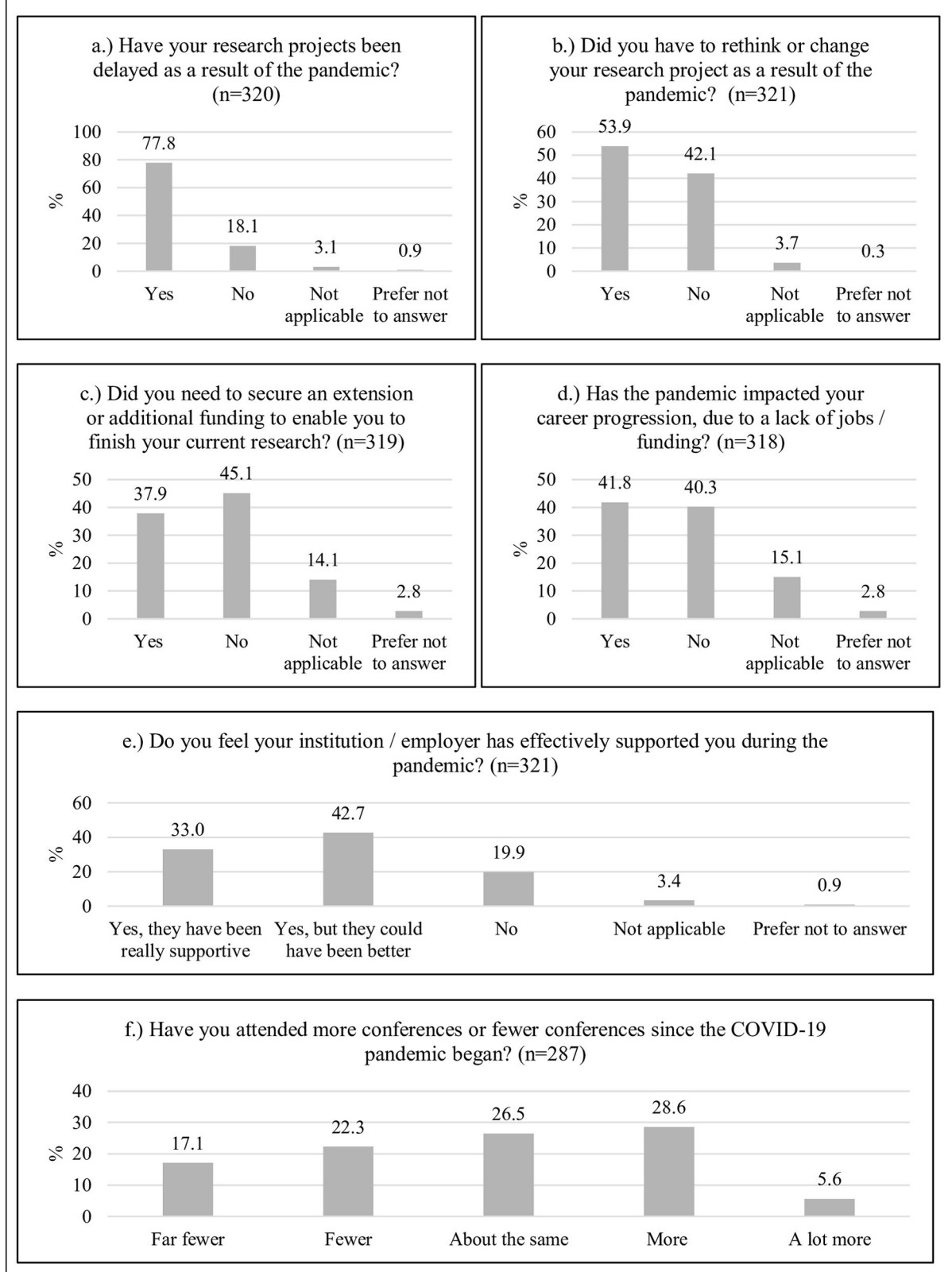

**Fig 1. Impact of the pandemic on surveyed ECDRs (in percentages).**

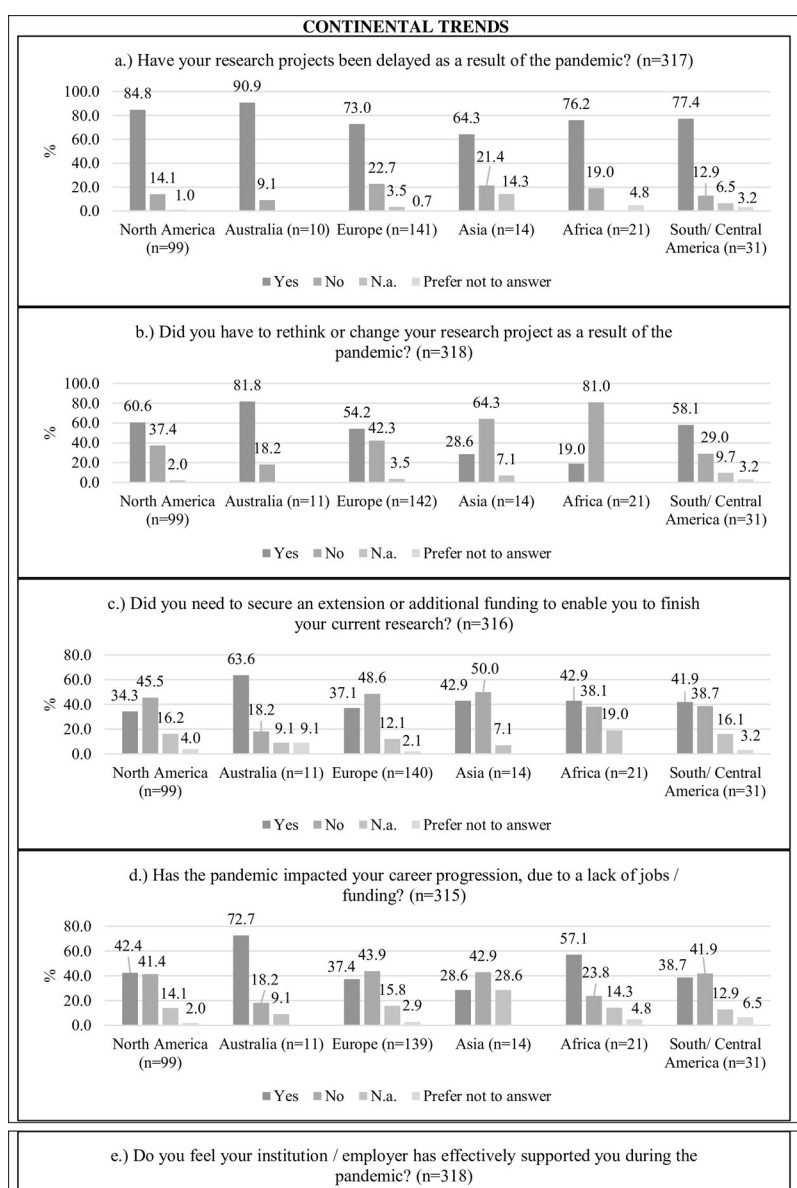

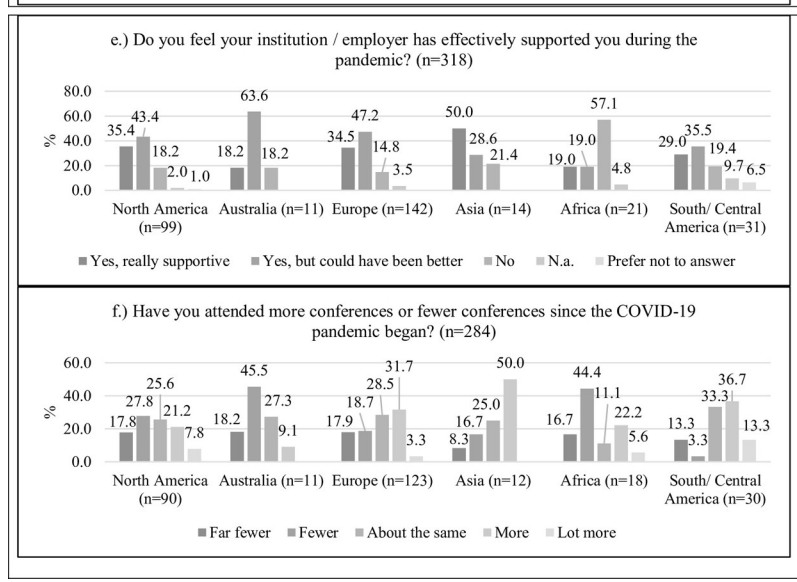

**Fig 2. Continental trends of the impact of the pandemic on surveyed ECDRs (in percentages).** Note: ECDRs who did not report a country were not include in these graphs.

There were no significant age differences for the following: need to rethink or change research projects, $X^2$ (df = 9, n = 320) = 5.81, p = .72, need to secure additional funding or an extension, $X^2$ (df = 9, n = 318) = 11.94, p = .22, the experienced impact on career progression, $X^2$ (df = 9, n = 317) = 6.13, p = .73, or the perceived support from the institution or employer, $X^2$ (df = 12, n = 320) = 12.96, p = .37. However, frequencies in age groups differed significantly with regards to the delays in ECDRs' research projects as a result of the COVID-19 pandemic, with the youngest group least likely to report delays, $X^2$ (df = 9, n = 319) = 18.32, p = .03 (Fig 3).

## Differences related to having dependents under 18 and being a primary caregiver

The frequencies of ECDRs who indicated having dependents under the age of 18 (n = 74) and without dependents under the age of 18 (n = 243) were compared. There were no significant differences between ECDRs with and without dependents under the age of 18 in the impact of the pandemic in any category: research project delays, $X^2$ (df = 3, n = 316) = 3.36, p = .34; changes or adjustments in research projects, $X^2$ (df = 2, n = 317) = 1.47, p = .48; the need to secure for additional funding or an extension, $X^2$ (df = 3, n = 315) = 4.59, p = .21; perceived impact on career progression, $X^2$ (df = 3, n = 314) = 2.47, p = .48; perceived support from institutions or employers, $X^2$ (df = 4, n = 317) = 5.1, p = .28; or conference attendance, $X^2$ (df = 4, n = 284) = 2.8, p = .59.

Furthermore, the frequencies of ECDRs replies who indicated being a primary caregiver (n = 69) or not being a primary caregiver (n = 247) were compared. There were no significant differences between ECDRs with and without caregiver responsibilities regarding the pandemic's impact in any category: research project delays, $X^2$ (df = 3, n = 315) = 2.73, p = .44; changes or adjustments in research projects, $X^2$ (df = 2, n = 31) = 1.17, p = .56; the need to secure for

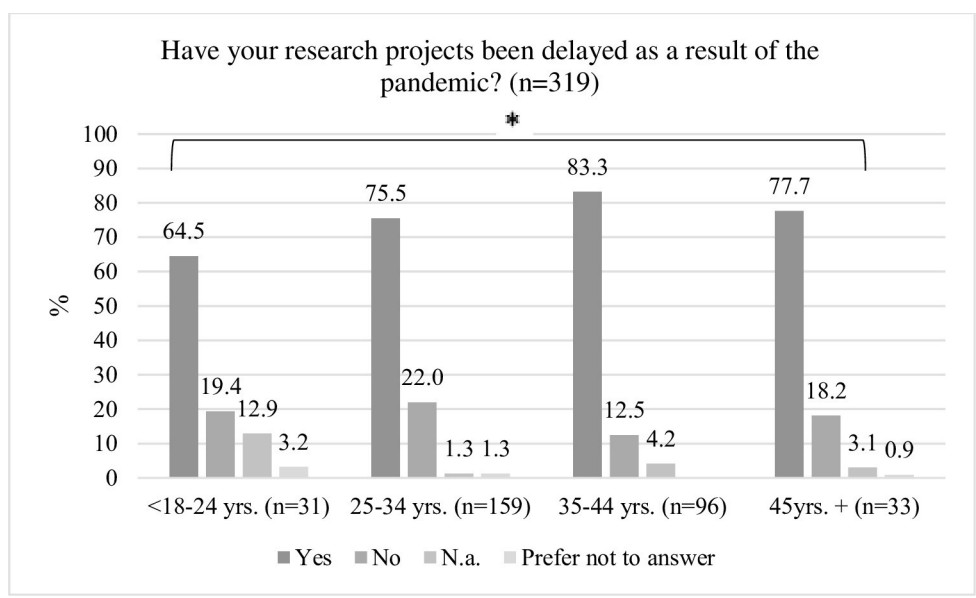

**Fig 3. Significant difference in age group frequencies (*p = 0.03).**

additional funding or an extension, $X^2$ (df = 3, n = 314) = 3.39, p = .50; perceived impact on career progression, $X^2$ (df = 3, n = 313) = 4.1, p = .25; perceived support from institutions or employers, $X^2$ (df = 4, n = 316) = 4.41, p = .35; or conference attendance, $X^2$ (df = 4, n = 282) = 5.19, p = .27.

## Discussion

This study shows that the COVID-19 pandemic has a substantial impact on ECDRs world-wide. Delays and adjustments of projects, the need for contract extensions and additional funding, and a general impact on career progression were reported. Only 1 in 3 ECDRs surveyed was fully satisfied with their institution's/ employer's support, while the majority expressed receiving either no support, or that institutions/employers was supportive, but could have done more. While the pandemic strained academics of all career stages, this expressed feeling of neglect underlines the necessity for the research community to increase efforts and support for ECDRs, as also highlighted in other fields [7, 18, 19]. A positive impact of the COVID-19 pandemic was the reported increase in conference attendance in ECDRs. Hybrid and online events enabled broader audiences to attend dementia-related events, which was also highlighted in other reports [20], and this topic is discussed further below.

No differences related to ECDRs' gender (men vs. women) were found in the present sample, and previous studies on the impact of the COVID-19 pandemic in relation to gender are inconclusive [4, 21, 22]. Evidence exists that women in dementia research get published less, receive less funding, and transition into advanced academic positions at disproportionally lower rates than men [20] and gender disparities for ECDRs require more attention prospectively. While ethnicity and socioeconomic background can also be relevant factors influencing career progression [23], no differences related to ECDRs' race (being vs. not being treated or perceived as a minority group or person of color) in the context of the pandemic were found in this sample. Moreover, age differences were only found in regard to reported delays in research projects, with ECDRs ≤24 years of age reporting delays less frequently compared to other the age groups. One possible explanation for this finding is that ECDRs of this age are more likely to be undergraduate students (vs. graduate students, postdocs, or faculty), and therefore are likely to have fewer direct research project leadership responsibilities which would be delayed by the pandemic. On the other hand, responsibilities outside of the workplace could also explain this result. Although ECDRs across the age-spectrum have family or other responsibilities outside of their research work, these types of responsibilities may be more highly concentrated among ECDRs >24 years of age. Having young children was found to be a strong predictor for research interruption during the pandemic, especially in women [22]. However, in the present sample, this age difference does not appear to be explained by having a dependent under the age of 18 (vs. not having a dependent) or being a primary caregiver (vs. not being a primary caregiver). Prospectively, it will be important to better understand the sources of this age-related difference in order to appropriately tailor supports to ECDRs, as this question was beyond the scope of the present study.

Continental trends, although not statistically analyzed due to small sample sizes, revealed that the impact of the pandemic on ECDRs varied around the world. This observation could be explained by the governmental and cultural reactions to the pandemic, which differed between countries. Furthermore, access to technologies to switch from face-to-face to remote data collection methods might not have been available to all ECDRs or participants, leading to negative consequences in some contexts. In the U.S., for instance, smartphone ownership increased in older adults (65+ years of age) from 46% in 2018 to 61% in 2021 [24], which is a positive direction for remote dementia research although this ability to participate remotely does not appear available to all older adults.

It is relevant to address the persisting impact of challenges resulting from the pandemic and highlight dementia- and continent-specific resources. The authors provide a list of such resources in S1 File and encourage colleagues to build on this overview. Below, we review insights gained through the pandemic and reactive strategies, building on Climie and Marques (8), may improve career development, progression, and support for ECDRs. Ideally, these insights and approaches learned during this time also serve as useful lessons to continue applying beyond the pandemic.

1. **Training, networking, and peer-support.** COVID restrictions and cancellations limited in-person training, events, and meetings, which usually present opportunities for ECDRs to network, gain new knowledge, and benefit from peer-support. However, digital solutions emerged quickly. Prospectively, institutions and employers are encouraged to provide online courses and peer-support meetings more often, and ECDRs may also explore options outside their own institution. Online networking events can be planned and included in courses or conferences, for example, by using break-out rooms. Moreover, social media platforms offer possibilities to connect with fellow researchers, and training focused on increasing confidence and mastering these forms of networking and outreach could be helpful.

2. **Data collection and publishing.** Quantity, including publication volume, and quality of doctoral students' performance are indicative of future excellence [25], and the pandemic certainly limited ECRs' work processes. To address delays with research projects and publications relevant for career progression, ECDRs might have adjusted research study recruitment strategies, such as using digital procedures, which can also be considered in the future. Whenever face-to-face interactions with participants are necessary but challenging to execute (due to the pandemic or other circumstances), alternatives for ECDRs could include collaborations with peers, literature reviews, or secondary analysis of existing data sets. New publication formats for ECDRs in the field of dementia are also helpful to promote publishing, for instance, through special issues featuring the work from early career scientists, as also suggested by Johnson and Weivoda [2]. Moreover, social media outlets, including Twitter, Facebook, LinkedIn, TikTok, or Instagram, represent ways to disseminate research findings. ECDRs can either share hyperlinks to publications or highlight preliminary results via these platforms to reach a wide audience, which can have an additional societal impact. There is an academia movement towards tracking this type of impact via Altmetrics [26].

3. **Attending and presenting at events.** Most in-person events were cancelled or postponed during the pandemic, robbing ECDRs of the exposure to and experience of presenting their work to a live audience and receiving direct feedback from fellow investigators. On one hand, digital conferences may limit direct interactions. On the other hand, employers should still encourage professional society online presentations and participation in online conferences to allow ECDRs to receive feedback and practice their communication skills. The present findings show that conference attendance generally increased in ECDRs since the start the pandemic, which can be seen as a positive impact. ECDRs can also practice presenting at digital/ hybrid research group meetings of their own or collaborating institutions. Presentations can be recorded and uploaded to institutional or personal websites to facilitate dissemination further and demonstrate ECDRs' expertise. ECDRs are encouraged to provide feedback to fellow researchers during digital events, as even a tweet or message in a chat can increase one's network. Furthermore, participating in online conferences is always substantially less expensive and time-consuming, and better for the climate than travelling

(e.g., by plane) to in-person events, which may explain the reported increase in conference attendance by European, Asian, and South/Central American ECDRs. In 2020, more women attended large dementia conferences [20], and online events might help ECDRs with children or other care or household responsibilities to combine work with personal tasks. Moving forward, hybrid events are likely more inclusive for ECDRs and facilitate international participation, specifically when project and training budgets are small. Online conference formats are thought to enhance sustainability, equity, and inclusivity [27] and the dementia field can benefit from this approach.

4. **Awards.** Awards and prizes are crucial for the ECDRs' curriculum vitae, promotions, and tenure processes, as any distinctions and merits can enhance career progression. The pandemic reduced chances for awards at in-person events, however, digital awards were included in some conferences, such as the 'Twitter award' at Alzheimer Europe conference in 2020 and 2021. Prospectively, institutional and professional society awards offer a possibility to apply for and receive recognition. ECDRs can also submit application for early career investigator awards, for instance, at scientific journals or networks. Employers and supervisors can support their ECDRs through reference letters or nominations, and in some instances, ECDRs can nominate one another or self-nominate, and this should be encouraged.

5. **Funding.** Changes in distribution of funding due to the urgent need for COVID-19 research might have reduced chances for dementia-specific funding to some extent. Therefore, grant options from industry, insurances, or private grants may be explored and ECDRs could also become co-applicants on larger grants. The ISTAART PEERs PIA is currently reviewing governmental and charity funders worldwide to promote available options (manuscript in preparation). One funder (the UK Alzheimer's Society) has proven that adjustments of ECDR contracts are possible, as they extended fellowships to four years during the pandemic, and other funders should follow this example. Moreover, fellowships and contracts should provide sufficient funds for ECDRs to cover living wages, which is currently not always the case. It is not uncommon for students to have a second job to cover their costs, potentially leading to increased stress and time restrictions for conducting dementia research.

6. **Promotion and tenure review requirements.** Research shows that organizations may attract, motivate, and retain employees by supporting their employees' career development [28], which is also of relevance in the context of this study. ECDRs have been unable to work in the typical ways during the pandemic. Researchers may have had trouble collecting data, including preliminary data (necessary to for grant writing), writing and publishing papers, and writing grant proposals and obtaining new funding, and therefore institutions and supervisors should adjust promotion and tenure review requirements to accommodate this fact; such an approach has been called for by colleagues [29]. Funders can further provide COVID-related funding extensions, additional financial supports, for instance for childcare, flexibility with preliminary data requirements, and the like. Many of these approaches were taken by the US National Institution of Health [30]. Finally, ECDRs themselves can consider including a "COVID-19 impact statement" in promotion applications, in which the individual impact is highlighted, such as reduced working hours due to caring duties during the pandemic or issues with recruitment due to lock-down.

7. **Improve ECDRs well-being.** The COVID-19 pandemic brought to light the strong overlapping influences of career and mental health on overall researcher well-being [31]. Increased levels of stress and anxiety during the pandemic can have a negative effect on

cognitive functioning and thus productivity, and researchers are encouraged to be understanding and patient with colleagues, students, and ourselves [32]. Supervisors, mentors, or managers should generally check-in on ECDRs, and stress-management courses can also be offered online. As it can be challenging to support an ECDR, particularly in times of crisis, supervisors and anyone providing help might in turn benefit from emotional guidance themselves. Being aware of one's mental health, engaging in coping strategies, such as exercises, meditation, mindfulness training, and sharing experiences with others may be helpful. In line with this suggestion are the perceived positive changes during lockdown reported by researchers, including increased outdoor activities and interacting with family or friends [4]. Furthermore, institutions are asked to adjust their expectations and requirements, such as number of publications necessary for a promotion or tenure, while providing additional support (e.g., caregiving) to reduce the stress on ECRs in the first place [29]. Finally, interindividual differences in preferences for working environment should be acknowledged moving past the pandemic [4] as part-time and home-office schedules may support a better work-life balance for ECDRs.

8. **Inter-sectoral collaborations and career transitioning.** Due to job-security related stress, job-related dissatisfaction, and salary, ECRs might consider a career change [33]. During especially uncertain times such as a global pandemic, this urge might grow. To enhance employability outside of one's own sector, inter-disciplinary, and inter-sectoral training is generally advisable, while collaborations with healthcare, industry, or policy can also advance dementia research [34]. Career counselling and creating lists of transferable skills represent further possibilities for ECDRs to explore alternative career paths.

## Limitations

Study limitations include an over-representation of participants from Central Europe (44.2%) and North America (30.8%), while only a small proportion of participants were based in Asia, Africa, Australia, South America, or South-East European countries. Although several channels were used to engage ECDRs around the world, recruitment bias is likely to still be present. Furthermore, participants self-identified as being in their early career, but associate/ full professors (4%) also joined this survey. Some countries might view these titles as mid- or advanced-stage research positions, potentially introducing a bias in the results. In the present study, the subjective feeling of being at an early career stage was of primary interest. If differences based on objective career stage and title rather than self-identified career stage are of interest in future work, data from such respondent could be removed from analysis or years of experience working in dementia research could be included as a control variable in multivariable modeling. Also, it was not assessed how the delayed ECDRs projects were funded; this information would have been useful to interpret the described impact in more detail. Finally, a more advanced statistical analysis was not suitable due to sample heterogeneity. Prospectively, additional lifestyle and personal factors could be explored, qualitative information might help to understand the individual context better, and a comparison to non-pandemic challenges would be interesting.

## Conclusion

The COVID-19 pandemic resulted in complex challenges for ECDRs worldwide. With increased attention and support, the future of dementia research may certainly benefit from the lessons-learned, especially in regard to hybrid and online events to enhance access to knowledge exchange. Echoing a growing call for action, research and academic institutions,

governments, funding agencies, and advanced experts in the field of dementia have the responsibility to improve support for ECDRs [16], and encourage and empower ECDRs to be proactive and ask for help when needed.

## Supporting information

**S1 File. A list of dementia-specific resources for Early Career Researchers collected by the ISTAART PEERs continental leads.**
(DOCX)

## Acknowledgments

The authors thank Dr. Claire Sexton, Jodi Titiner, and the Alzheimer's Association International Society to Advance Alzheimer's Research and Treatment (ISTAART) for their support. We acknowledge also the conversations with Dr Deborah Oliveira and Dr Wade Self, who helped to shape the survey. Thanks goes to all researchers that beta-tested and completed this survey, and shared it with their networks. Furthermore, we are grateful for the input from the members of the Professional Interest Area to Elevate Early Career Researchers (PEERs PIA), namely Dr. James Quinn (Programs Chair), Dr. Naiara Demnitz (Communications Chair), Dr. Tengfei Guo (Continental Lead-Asia), and Dr. Diana Karamacoska (Continental Lead-Australia). Next to these four members, the authors SLB (Continental Lead-Europe), CES (Vice Chair), WSB (Continental Lead-South America), LAW (Continental Lead-North America), RF (Continental Lead-Africa), and AS (Chair) were part of the ISTAART PEERS PIA. CES is the current chair and questions related to the ISTAART PEERS PIA can be directed to her: Beth.Shaaban@pitt.edu.

## Author Contributions

**Conceptualization:** Sara Laureen Bartels, C. Elizabeth Shaaban, Wagner S. Brum, Lindsay A. Welikovitch, Royhaan Folarin, Adam Smith.

**Data curation:** Sara Laureen Bartels.

**Formal analysis:** Sara Laureen Bartels, C. Elizabeth Shaaban.

**Methodology:** Sara Laureen Bartels, C. Elizabeth Shaaban, Adam Smith.

**Project administration:** Adam Smith.

**Resources:** Adam Smith.

**Software:** Adam Smith.

**Visualization:** Sara Laureen Bartels.

**Writing – original draft:** Sara Laureen Bartels.

**Writing – review & editing:** C. Elizabeth Shaaban, Wagner S. Brum, Lindsay A. Welikovitch, Royhaan Folarin, Adam Smith.

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
