## [Decision Letter · Decision Letter 0]

4 Sep 2022

PONE-D-22-18562Impact of the COVID-19 pandemic on early career dementia researchers: A global online surveyPLOS ONE

Dear Dr. Bartels,

Thank you for submitting your manuscript to PLOS ONE. After careful consideration, we feel that it has merit but does not fully meet PLOS ONE’s publication criteria as it currently stands. Therefore, we invite you to submit a revised version of the manuscript that addresses the points raised during the review process.

We look forward to receiving your revised manuscript.

Kind regards,

Maria Elisabeth Johanna Zalm, Ph.D

Editorial Office

PLOS ONE

Journal Requirements:

a) Did participants provide their written or verbal informed consent to participate in this study?

 "No funding was received for this work specifically. CES was funded by grant T32AG055381 from the National Institute on Aging of the United States National Institutes of Health. AS is funded by the National Institute for Health Research through the NIHR University College London Hospitals Biomedical Research Centre." 

5. We noted in your submission details that a portion of your manuscript may have been presented or published elsewhere. "A summary report of the complete survey (raw data, non peer-reviewed) is also published under Smith A, Shaaban CE, Bartels SL, Welikovitch L, Brum W, Folarin R. Listening to Early Career Researchers [Report]. 2022 [Available from: " ext-link-type="uri" xlink:type="simple">https://www.dementiaresearcher.nihr.ac.uk/survey." Please clarify whether this [conference proceeding or publication] was peer-reviewed and formally published. If this work was previously peer-reviewed and published, in the cover letter please provide the reason that this work does not constitute dual publication and should be included in the current manuscript.

7. We note that you have stated that you will provide repository information for your data at acceptance. Should your manuscript be accepted for publication, we will hold it until you provide the relevant accession numbers or DOIs necessary to access your data. If you wish to make changes to your Data Availability statement, please describe these changes in your cover letter and we will update your Data Availability statement to reflect the information you provide.

8. One of the noted authors is a group or consortium "ISTAART PIA". In addition to naming the author group, please list the individual authors and affiliations within this group in the acknowledgments section of your manuscript. Please also indicate clearly a lead author for this group along with a contact email address.

Additional Editor Comments:

Your manuscript has been assessed by one peer-reviewer and their report is appended below. 

The reviewer comments that the manuscript could be strengthened by providing more detail to certain aspects of the study, including the introduction, methodology, and discussion section. In addition, the reviewer states that the results should be interpreted more cautiously, and the limitations section requires more attention. 

Please note that we have only been able to secure a single reviewer to assess your manuscript. We are issuing a decision on your manuscript at this point to prevent further delays in the evaluation of your manuscript. Please be aware that the editor who handles your revised manuscript might find it necessary to invite additional reviewers to assess this work once the revised manuscript is submitted. However, we will aim to proceed on the basis of this single review if possible. 

Reviewers' comments:

Reviewer's Responses to Questions

**Comments to the Author**

1. Is the manuscript technically sound, and do the data support the conclusions?

Reviewer #1: Yes

2. Has the statistical analysis been performed appropriately and rigorously? 

Reviewer #1: Yes

3. Have the authors made all data underlying the findings in their manuscript fully available?

Reviewer #1: Yes

4. Is the manuscript presented in an intelligible fashion and written in standard English?

Reviewer #1: Yes

5. Review Comments to the Author

Reviewer #1: This is an interesting paper worthy of publication as it focuses on the impact on early career researchers. It is well written and has some useful recommendations at the end of the paper. I have some suggestions for strengthening the paper.

Abstract- I do think the results could include the findings about conference attendance as the change to remote delivery has been one of the few positive impacts of the pandemic. The conclusion section could be re-written to provide stronger conclusion to the abstract, rather than writing a list of content try instead to focus on the main messages from the paper.

Introduction (page 3) I would advise including a definition of ECR given this is the focus of the paper.

Introduction (page 3) The first paragraph could make more of why this is an important area to look at. I appreciate the need to understand the experiences of dementia ECRs, but this paper will be read by those from different disciplines.

Introduction (page 3) it wasn’t clear if all the studies you were reviewing were specifically conducted to explore the impact of the pandemic on researchers.

Introduction (page 4) you discuss important studies that were conducted despite the pandemic. Would be worth a reference to those studies that were conducted to explore the impact of the pandemic on people with dementia and carers, these happened despite all the challenges around recruitment.

Method (page 6) please provide more details of the questionnaire topics, this could be in the form of a supplementary document.

Method (page 6) it would be useful to contextualise the time period data collection period in relation to the pandemic

Results (page 9) “delay I their projects” is that their own projects (e.g. PhD project) or the PIs project? Or is that not clear in the data. Delays have different implications depending on roles within the project.

Results (page 9/10) I would advise I more cautious interpretation of the findings. You state nearly two-thirds felt at least somewhat unsupported but the data shows only 19.9% felt unsupported. Whilst, 41.2% felt employers were generally supported but could do better, I don’t see this data as implying they were somewhat unsupported because they said they were generally supported. I would apply this comment to how you have reported the results in the abstract and on page 13 of the discussion section where you say only 1 in 3 ECRs were satisfied with the support.

Discussion (page 13) with the discussion of the gender differences in the data it might be worth a reference to the “dementia research career pipeline” 2022 paper which looked at gender differences.

Discussion- before you get to discussing the lessons learnt it is worth some discussion of the finding about conference attendance.

Discussion (page 14) you mention continental trends with regards to reactions to the pandemic, worth considering issues around access to technologies both for researchers and participants e.g. for remote interviewing.

Discussion (Page 15) Data collection and publishing, I think there are other ways for ECRs getting findings out there and building their profile e.g. through twitter or tik tok.

Discussion (Page 16)- A couple of issues to consider for funding. One for funders providing funding for longer periods e.g. UK Alz Soc recently changed to funding 4 years of a PhD as recognised students do need longer than 3 years. Second funding to be sufficient for people to be paid living wages e.g. with the current crisis PhD stipends are not sufficient and many PhD students are taking on second jobs.

Discussion (Page 17), promotion, again this is just my opinion, but one thing is to include Covid-19 impact statements in promotion applications so people can talk about the impact on them e.g. if they had to reduce hours for caring duties during the pandemic.

Discussion (Page 18)- improve ECRs well-being. Support for ECRs should also include support for line managers and PIs, emotionally supporting ECRs can take its toll.

Discussion (page 18) limitations. You need to acknowledge that sample self-defined as ECRs so it looks like you have some associate prof/ Profs in the sample with aren’t typically seen as ECRs. The study didn’t specifically include mid-career dementia researchers. By using questionnaire methods it meant that pertinent issues couldn’t be followed up with participants as could be done with an interview e.g. clearly some people were unable to finish their research- why was that?

6. PLOS authors have the option to publish the peer review history of their article (what does this mean?). If published, this will include your full peer review and any attached files.

Reviewer #1: No

---

## [Author Response · Author response to Decision Letter 0]

18 Oct 2022

Dear reviewer, 

thank you so much for taking the time to review our manuscript. We appreciate your input, questions, and remarks as they strengthen the article. Please see our detailed reply in the response letter, where we elaborate on each point individually. 

Kind regards,

On behalf of all authors, 

Dr Sara Laureen Bartels

---

## [Decision Letter · Decision Letter 1]

28 Oct 2022

Impact of the COVID-19 pandemic on early career dementia researchers: A global online survey

PONE-D-22-18562R1

Dear Dr. Bartels

We’re pleased to inform you that your manuscript has been judged scientifically suitable for publication and will be formally accepted for publication once it meets all outstanding technical requirements.

Kind regards,

Gabriel G. De La Torre

Academic Editor

PLOS ONE

Additional Editor Comments (optional):

I consider that authors have improved the article with this revision and despite some concerns regarding characteristics of the sample (mostly women), the manuscript in its present form can be accepted for publication.

---

## [Editor Report · Acceptance letter]

1 Nov 2022

PONE-D-22-18562R1 

Impact of the COVID-19 pandemic on early career dementia researchers: A global online survey 

Dear Dr. Bartels:

I'm pleased to inform you that your manuscript has been deemed suitable for publication in PLOS ONE. Congratulations! Your manuscript is now with our production department. 

Kind regards, 

on behalf of

Dr. Gabriel G. De La Torre 

Academic Editor

PLOS ONE